# A Rapid Quantitative Analysis of Bicomponent Fibers Based on Cross-Sectional In-Situ Observation

**DOI:** 10.3390/polym15040842

**Published:** 2023-02-08

**Authors:** Jieyao Qin, Mingxi Lu, Bin Li, Xiaorui Li, Guangming You, Linjian Tan, Yikui Zhai, Meilin Huang, Yingzhu Wu

**Affiliations:** 1School of Textile Materials and Engineering, Wuyi University, Jiangmen 529020, China; 2College of Innovation and Entrepreneurship, Wuyi University, Jiangmen 529020, China; 3College of Intelligent Manufacturing, Wuyi University, Jiangmen 529020, China

**Keywords:** bicomponent fibers, quantitative analysis, melting and dissolving, cross-sectional in-situ observation, image processing, artificial intelligence (AI)

## Abstract

To accelerate the industrialization of bicomponent fibers, fiber-based flexible devices, and other technical fibers and to protect the property rights of inventors, it is necessary to develop fast, economical, and easy-to-test methods to provide some guidance for formulating relevant testing standards. A quantitative method based on cross-sectional in-situ observation and image processing was developed in this study. First, the cross-sections of the fibers were rapidly prepared by the non-embedding method. Then, transmission and reflection metallographic microscopes were used for in-situ observation and to capture the cross-section images of fibers. This in-situ observation allows for the rapid identification of the type and spatial distribution structure of the bicomponent fiber. Finally, the mass percentage content of each component was calculated rapidly by AI software according to its density, cross-section area, and total test samples of each component. By comparing the ultra-depth of field microscope, differential scanning calorimetry (DSC), and chemical dissolution method, the quantitative analysis was fast, accurate, economical, simple to operate, energy-saving, and environmentally friendly. This method will be widely used in the intelligent qualitative identification and quantitative analysis of bicomponent fibers, fiber-based flexible devices, and blended textiles.

## 1. Introduction

Bicomponent fibers are widely used in functional protective textiles, filter materials, and biomedical materials owing to their unique elastic, fluffy, multifunctional features and super-clean properties [1,2,3,4,5,6,7,8]. In addition, many scientists have focused on various fiber-based flexible devices. These new devices have broad application prospects in the fields of real-time healthcare and soldier communications [9,10,11,12]. The material, morphology, and distribution uniformity of each component in these electronic fibers and the bicomponent obviously affect their properties. A series of testing standards should be developed for these products prior to they being widely applied [13,14,15]. A good testing standard has the following characteristics: (1) the test results are stable and comparable among enterprises; (2) the detection equipment is economical, and the detection cost is low; (3) the operation steps are simple and suitable for ordinary detection technicians. Although there are some product standards for bicomponent fibers, there are no standards for the qualitative and quantitative analysis of each component and their mass percentage [16,17,18,19]. Compared with the fibrous longitudinal observation, the cross-sectional observation allows for the comprehensive and accurate characterization of the morphology and distribution of each fibrous component. Presently, the most commonly used method for observing fibrous cross-section is the resin embedding method. This method has the following disadvantages: cumbersome operation, a low success rate of preparing ultrathin cross sections, inability to implement in-situ dissolution and in-situ melting observation, and difficulty of application in future rapid intelligent detection [20,21].

Many high-level studies have been reported in recent years to solve the problem of making fibrous cross-section slices and intelligent identification. Binjie Xin’s research team used the traditional method of resin embedding to fix the ramie fiber bundles and proposed an edge-enhanced image processing technology to improve the identification accuracy of fiber edges [22]. Later, they gained the cross-sectional images of polyester/cotton blended fibers by placing fibers into a suction tube, injecting some prepared resin to embed fibers, and slicing with a microtome [23]. In 2022, they used a convolutional neural network to identify longitudinally overlapped wool/cashmere fibers [24]. The identification of polyethylene terephthalate (PET) and polytrimethylene terephthalate (PTT) fibers and the uncertainty analysis of the content determination of cotton/linen blended products have been completed by our team. To solve the problem of quantitative analysis of bicomponent fibers, an in-situ observation technology for non-embedded fibrous cross-section was developed in this work. Unlike the resin embedding method mentioned earlier, the non-embedded method was that the fibers were fixed by friction between crowded fibers in a narrow rectangular hole instead of solidified resin, cut smoothly to be captured images directly instead of cut into thin slices with a thickness of several micrometers.

## 2. Experimental Section

### 2.1. Reagents and Materials

Collodion (solution), formic acid, and basic sodium hypochlorite were purchased from the Tianjin Damao Chemical Reagent Factory. Core-sheath, side-by-side, lobe-type, and fibril matrix-type bicomponent fibers were purchased from SINOPEC Yizheng Chemical Fiber Co., Ltd. (Yangzhou, China).

### 2.2. Instruments

A slicer (W110 slicer, Jiangmen Kuafu Nanometer Instrument Research Institute Co., Ltd., Jiangmen, China) was used to clamp short fiber bundles. Two metallographic microscopes (CMY310, Beijing Shiji Kexin Instrument Co., Ltd., Beijing, China; CFNY-A1, Jiangmen Kuafu Nano-Instrument Research Institute Co., Ltd., Jiangmen, China) were used to capture images of in-situ observations. Figure 1, Figure 2c,d and Figure 3e–h were captured by CMY310; Figure 2a,b, Figure 3a–d and Figure 8 were captured by CFNY-A1. The melting point meter (PRT-8A, Beijing Shiji Kexin Instrument Co., Ltd., Beijing, China) was used to carry out in situ melting observation. The differential scanning calorimetry (DSC214, NETZSCH, Selb, Germany) was used to verify the melting point of bicomponent fibers. A metal 3D printer (EOS M3100, Tolerance, Balingen, Germany) was used to fabricate a new slicer. A super depth-of-field stereo microscope (VHX-7000, KEYENCE, Osaka, Japan) was also used to verify the area of the fibrous cross-section in this study.

## 3. Results and Discussion

### 3.1. In-Situ Qualitative Analysis

#### 3.1.1. Preparation of Short Fiber Bundles Cross-Sections

As shown in Figure 1a, after the slicer clamped the fiber bundle, the fibers exposed at the upper and lower ends of the slicer were cut. The slicer and fiber assembly were moved to a metallographic microscope. The in situ observations were carried out. Compared with the traditional resin embedding method, the operation was very easy for a general technician. It only took 10 min. It is very friendly to a new person. They can make good short fiber bundles cross-sections after half an hour of training. The fibrous surface texture is clear. There is a large gap between fibers in Figure 1b. These voids were conducive to mass and heat transfer during melting or dissolution. As shown in Figure 1c, the embedding agent embedded the voids between the fibers, which is not conducive to mass and heat transfer. As shown in Figure 1d, the polyester fibers were brighter than cotton fibers. The characteristic difference shows that synthetic and natural fibers can be quickly identified. This method can also be used to visualize the mixing uniformity of the blended fibers. Because the surface texture structure of all fibers could not be seen with the transmission observation method, the later identification experiment of bicomponent fibers was carried out using the coaxial reflection observation method.

#### 3.1.2. In-Situ Dissolving Test

A small amount of formic acid was injected into the short fiber bundles through the micropores of the slicer, and formic acid quickly diffused to each fiber through the capillary effect (as shown in Appendix A). As shown in Figure 2a,b, formic acid rapidly dissolved the small components connecting each large segment, and the large components remained the same (the video of the in-situ dissolution process can be seen in Appendix A). It can be determined that the small components were polyamide fibers, such as PA66, PA6, or PA11. A further melting point test was required to ensure an accurate composition.

As shown in Figure 2c,d, formic acid rapidly dissolved the sea or small component, leaving the round superfine fiber (island component) or large component. It can be determined that the small component was polyamide polymer, and may be PA66, PA6, or PA11. A further melting point test was required to ensure that the composition was accurate. Core sheath and side-by-side bicomponent fibers were required to obtain the comprehensive advantages of the two materials, which require good compatibility and are not easy to separate. Presently, bicomponent fibers are mainly composed of polyesters, polyolefins, or polyester/polyolefins. These fibers were insoluble in formic acid.

In addition, the method can be used for the rapid qualitative analysis of conventional blended products. It was more accurate than the traditional chemical dissolution qualitative method, such as the rapid dissolution identification of wool and acrylic fiber blended fabric shown in Appendix A.

#### 3.1.3. In Situ Melting Test of Bicomponent Fibers

It can be observed from Figure 3a–h that when the average temperature reached 135 ℃ and 170 °C, the sheath polymer and core layer were melted, respectively. The melting point of each type of bicomponent fiber was tested three times. The video of the melting process was seen in Appendix A. According to ISO standard “ISO TR 11,827 Textiles—Composition testing—Identification of fibres”, the sheath layer and core layer were determined to be the PE layer and PP layer with a density of 0.95 g/cm^3^ and 0.91 g/cm^3^. As shown in Figure 3c,d, when the average temperature reached 229 °C and 256 °C, two components were melted, respectively. According to ISO TR 11,827 and previous dissolution results, they were determined to be PTT polymer and PET polymer with a density of 1.33 g/cm^3^ and 1.38 g/cm^3^. As shown in Appendix A, in comparison with the longitudinal morphology melting of the bicomponent fiber, the melting component and its spatial distribution can be clearly observed in the fiber cross-section during the in situ melting process. Similarly, as shown in Figure 3e–h, when the average temperature reached 220 °C to 225 °C, the small component was melted and dissolved in formic acid. Thus, the small component was determined to be PA6 with a density of 1.13 g/cm^3^. Because the melting point of a large component was from 250 °C to 260 °C and this component was insoluble in formic acid, the large component was determined to be PET with a density of 1.38 g/cm^3^. The melting process video can be seen in Appendix A. It was more accurate and faster than the traditional melting point method. The standard deviations of the melting point were 2.1 °C to 2.8 °C. As the melting point increased, the standard deviation of the results increased. The thermal insulation performance of the heating table was not good enough. As a result, the test results fluctuated greatly.

It can be seen that the fiber in Figure 4a has two melting points: 134 °C and 171 °C. This fiber comprised PE and PP according to the melting point of conventional spinning materials. This result validates previous results of melting. The fiber in Figure 4b has two melting points: 227 °C and 253 °C. According to the melting point of conventional spinning materials and the dissolution test results of the fiber insoluble in formic acid, it can be seen that this fiber was composed of PET and PTT. This result was consistent with previous results of melting. The fiber in Figure 4c has two melting points: 225 °C and 255 °C. According to the melting point of conventional spinning materials and the experimental results obtained when this fiber was partially dissolved in formic acid, it can be seen that the fiber was composed of PET and PA6. The fiber in Figure 4d has two melting points, 218 °C and 254 °C. According to the melting point of conventional spinning materials and the experimental results that the fiber was partially dissolved in formic acid, it can be seen that this fiber was composed of PET and PA6. As the bulk density of the fibril matrix and the lobe bicomponent fiber were higher than that of core-sheath and side-by-side bicomponent fiber, resulting in a slower heat transfer rate and higher temperature of the heating plate to ensure all of the fibers clamped in the middle reached the melting point.

In addition to the aforementioned dissolution and melting, polarization can also be used to quickly identify bicomponent fibers and blended fabrics. As shown in Appendix A, side-by-side PET/PTT bicomponent fibers and wool/acrylic fiber blended products can be quickly identified by the polarization performance of the cross section.

### 3.2. Determination of the Mass Percentage for Each Component by AI Model

#### 3.2.1. Calculation of the Mass Percentage of Each Component

The mass percentage of each component was calculated based on the fiber volume, density, and test specimens of each component. Because the bicomponent fiber was a filament, the cutting length *L* of each component was equal and could be reduced mutually. Finally, the formula for calculating the mass percentage of each component was simplified as a function of the area ratio and density of each component, as shown in Formula (1).
(1)$$\begin{array}{l} X_{1i}=\frac{\rho_{1}V_{1}}{\rho_{1}V_{1}+\rho_{2}V_{2}}\times 100\%\\ =\frac{\rho_{1}S_{1}L_{1}}{\rho_{1}S_{1}L_{1}+\rho_{2}S_{2}L_{2}}\times 100\%\\ =\frac{\rho_{1}}{\rho_{1}+a_{21i}\rho_{2}}\times 100\% \end{array},$$
(2)$$\overset{-}{X_{1}}=\frac{\sum X_{1i}}{N},$$
(3)$$X_{2}=100-\overset{-}{X_{1}},$$
(4)$$a_{21i}=\frac{S_{2i}}{S_{1i}},$$
where $X_{1i}$ is the mass percentage of component 1 of i fiber, %; $\overset{-}{X_{1}}$ is the average mass percentage of component 1 in all test specimens, %; N is the total number of fiber specimens; $\rho_{1}$ is the density of component 1 in grams per cubic centimeter (g/cm^3^); $\rho_{2}$ is the density of component 2 in grams per cubic centimeter (g/cm^3^); $X_{2}$ is the mass percentage of component 2 in all test specimens, %; $a_{21i}$ is the ratio of the cross-sectional area of component 2 to component 1 of the fiber; $S_{1i}$ is the area of component 1 of fiber i in square micrometers (μm^2^); and $S_{2i}$ is the area of component 2 of fiber i in square micrometers (μm^2^).

The density of each component in the bicomponent fiber was determined after identification. The area of each component was measured using a CCD digital camera equipped with a microscope and AI software.

The number of test samples *N* is affected by three factors: the *CV* value of the test results, the coefficient t at a certain confidence level, and the allowable error rate *E*. The relationship between the three factors is shown in Equations (5) and (6):(5)$$E=t\cdot \frac{CV}{\sqrt{N}},$$
(6)$$N={\left(t\cdot \frac{CV}{E}\right)}^{2},$$

*E* is the allowable error rate, also known as the half-width value of confidence interval, %; the *t*-coefficient is achieved when the confidence level is 95%; *CV* is the coefficient of variation, %; and *N* is the number of testing specimens.

The *CV* value of the coefficient of variation can be obtained from the trend graph that changes with the number of testing specimens. This study focused on the analysis of the mass percentage of each component in the side-by-side and core-sheath bicomponent fibers. First, the cross-section, characteristic value, and mass percentage of each component in 100 fibers were collected and determined. Six data groups were randomly selected, and the number of fiber specimens in each group was 6, 9, 12, 18, 24, and 27, respectively. Five samples were taken randomly from each group, the average of the *CV* value of the fiber mass percent content was calculated from each group of five samples, and then the *CV* value change pattern was plotted for the testing specimens, as shown in Figure 5. When testing twelve side-by-side bicomponent fibers, the *CV* value was stable, gradually decreasing from 12% to 10%.

The variation rules of the *CV* under different magnifications were tested to investigate the influence of magnification on the reproducibility and stability of the test results. It can be seen from Figure 6 that the magnifications were ’600, ´1200, and ´2400. The greater the magnification, the higher the reproducibility of the test results. With an increase in the number of test specimens, the stability and reproducibility of test results obtained at ´1200 and ´2400 tended to be consistent, reaching 11–10%. To increase the number of fiber specimens in each image and improve the inspection efficiency, we recommend selecting a magnification of ´1200 to collect images. Considering other influencing factors and increasing the stability of the test results, the *CV* value of the test results of the side-by-side bicomponent fiber was selected as 11%.

The core-sheath bicomponent fiber was tested in the same manner. As the percentage of each component of the core-sheath fiber was significantly different during production, its distribution in the core or sheath was not as uniform as that of the side-by-side bicomponent fiber. The *CV* value was relatively larger, which was stable between 0.14 and 0.13, and should be selected as 14%.

The diameter uniformity of the bicomponent fiber was determined by its production technology and management level and was constantly improving. The relevant standards for bicomponent fibers are listed in Table 1. If the allowable deviation rate *E* was selected as ±4% or smaller, that will promote the improvement of the production technology, and these products will reach first grade or premium grade. Thus, the allowable error rate *E* was selected as 4% in this work according to FZ/T 52037-2014 in Table 1.

The confidence level for the calculation was 95%. According to GB/T 14335-2008 (The complete data is shown in Table A1 in Appendix B), the functional relationship between *N* and *t* is listed in Table 2.

Finally, the calculation of the testing specimens:Testing the number of side-by-side bicomponent fibers (7).
(7)$$N={\left(t\cdot \frac{CV}{E}\right)}^{2}=(2.03\times \frac{11\%}{4\%}{)}^{2}=31.2\approx 32,$$


2.The number of core-sheath bicomponent fibers was tested (8).

(8)
$$N={\left(t\cdot \frac{CV}{E}\right)}^{2}=(2.01\times \frac{14\%}{4\%}{)}^{2}=49.49\approx 50,$$



#### 3.2.2. Establishment of the Identification Model of Each Component Based on the Cross Section

In the previous section, the calculation methods for testing specimen *N* and the mass percentage content of each component were described. The cross-sectional area of each component must also be determined according to the calculation formula for the mass percentage content of each component. Image processing technology should be applied to establish an edge recognition model to quickly obtain an accurate cross-sectional area of each component. The edge recognition model should be suitable for transmission and reflection images of a variety of bicomponent fibers. Thus, the accuracy and efficiency of obtaining each component fiber’s cross-sectional area and characteristic values can be improved.

The process and method for establishing an image recognition model are described as follows: The image recognition model is primarily based on the YOLO V5 algorithm. The multiscale and multidimensional feature vectors of the labeled image were extracted using the convolution network. Subsequently, the positioning and classification of the fiber cross-section were completed. As shown in Figure 7, the DeeplabV3 + model was first established, and then the labeled fiber cross sections were input into the model for training. The encoder body of the model was a DCNN with void convolution. The most commonly used classification network for this model is ResNet. The model has a spatial pyramid pooling module with cavity convolution that can introduce multiscale information. With a gradual increase in the number of network layers of the training model, the feature information of the target was gradually enhanced, and the prediction ability of the target was also improved.

To reduce the cost to potential customers in the future, this study selected two types of medium-class metallographic stereomicroscopes commonly used in the industry as the data acquisition platform. In this work, on these cost-effective metallographic stereomicroscopes, the transmission and reflection images of cross sections of various bicomponent fibers were obtained using the non-embedding method with ´1200 magnification. The labeled images of these images were then drawn and used to train the recognition model. These fibers included 200 core-sheath bicomponent fibers, 200 side-by-side bicomponent fibers, 200 fibril matrix bicomponent fibers, and 200 lobe bicomponent fibers.

To expand the application of metallographic stereoscopic microscope and image recognition software in the quantitative analysis of blended fabrics, after the magnification was set to ´1200, the cross-section transmission images and reflection images of more than 10,000 fibers were collected, respectively. All fiber images were labeled to train the image recognition models. For example, a labeled image of the viscose fibers is shown in Appendix A.

After the model was established, its identification accuracy was verified. At ´1200 magnification, 50 fiber cross-section reflection images were collected, and the edge and area of the reference images were obtained and determined by manual drawing. The difference between them was within 2%, and the accuracy was more than 98%, which achieved the expected effect and could be used for the actual inspection. After qualitative analysis, the quantitative analysis based on image processing only took 10 min. It was faster than the traditional chemical dissolution method. The latter should take more than two hours and consume 100 mL of solvent in each sample.

#### 3.2.3. Application of Identification Model of Each Component on Cross Section

The verified software was used to process the image of the bicomponent fiber and obtain the cross-sectional area of each component in the fibers. As shown in Figure 8a–d, the software identified the real effect of the cortex and core layer. The morphology and area of each component were extracted accurately from the original image. The video of identifying the fiber edge with the image recognition model can be seen in Appendix A. The mass percentages of polyethylene (PE) and polypropylene (PP) in the core-sheath bicomponent fiber were 57.4% and 42.6%, respectively, and the *CV* value of the coefficient of variation was 14.3%. The deviation from the nominal value of the manufacturing enterprise was 2.6%, and the error in the results was less than 3%, meeting the inspection requirements. Similarly, at least 32 fibers should be tested as side-by-side bicomponent fibers. In this bicomponent fiber, the mass percentage of polyethylene terephthalate (PET) was 58.3%, the mass percentage of propylene terephthalate (PTT) was 41.7%, and the *CV* value was 8.4%. The deviation from the manufacturing enterprise’s nominal value was 1.7%, and the error in the result was less than 3%, which meets the inspection requirements. This result was also consistent with the ultra-depth-of-field microscope testing results.

Compared with the testing results of the chemical dissolution method, the testing results of the fibril matrix and lobe bicomponent fibers differ by 2%, which indicates that the microprojection method is not only reliable but also energy-saving, safer, faster, and more environmentally friendly than traditional chemical dissolution method.

## 4. Conclusions

Through in situ reflection observation of the non-embedded fiber bundles cross-section, each component material and its spatial distribution can be accurately identified on the cross-section of the bicomponent fiber, and the density (ρ) of each component can be determined. Cross-sectional transmission and reflection images of bicomponent fibers and conventional textile fibers were collected, labeled, and applied to establish image edge recognition models. Edge recognition models were verified to be accurate and reliable, with an accuracy rate of more than 98%. The cross-sectional profile of each component in the bicomponent fiber was accurately identified using this image recognition model, and the corresponding area, S, was calculated. The total number of fibers (*N*) was determined to be 30–50 by the *CV* value, confidence coefficient *t*, and allowable error rate *E*. Finally, the mass percent content of each component in the bicomponent fiber was calculated. Comparing its production nominal value and chemical dissolution method, the results were accurate and reliable, with a deviation of less than 2%. This method provides an important reference for formulating a quantitative testing standard for bicomponent fibers and fiber-based flexible devices.

## Figures and Tables

**Figure 1 polymers-15-00842-f001:**
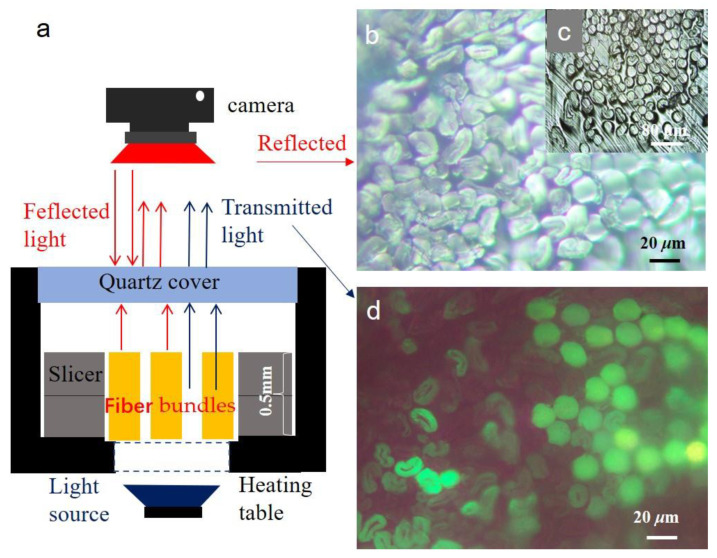
Transmission and reflection observation diagram of the cross-section of polyester/cotton blended fibers (**a**) test schematic illustration, (**b**) reflection observation diagram of polyester/cotton blended fiber non-embedded short fiber bundles, (**c**) reflection observation diagram of polyester/cotton blended fibers embedded slice, (**d**) transmission observation diagram of polyester/cotton blended fibers non-embedded short fiber bundles.

**Figure 2 polymers-15-00842-f002:**
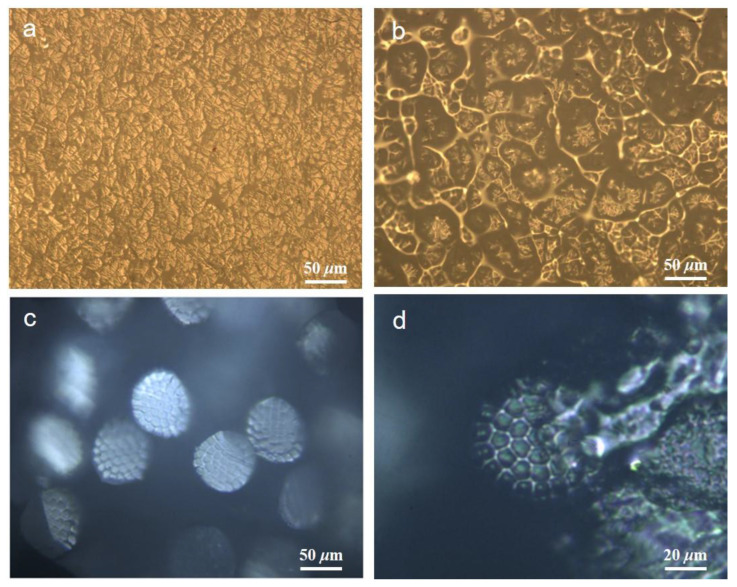
In situ dissolution diagram of lobe type and fibril matrix bicomponent fibers (**a**) before the dissolution of lobe type bicomponent fibers (**b**) after the dissolution of lobe type bicomponent fibers (**c**) before the dissolution of fibril matrix bicomponent fibers (**d**) after the dissolution of fibril matrix bicomponent fibers.

**Figure 3 polymers-15-00842-f003:**
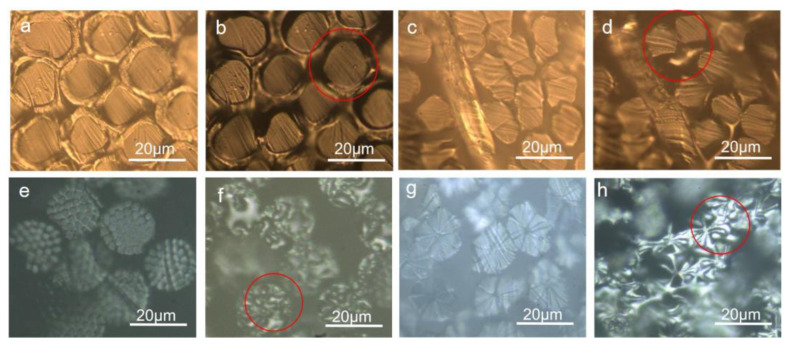
Before and after in-situ melting of core-shell, side-by-side, fibril matrix, and lobe bicomponent fibers. Magnification: 2400, (**a**,**b**) before and after melting of the PE/PP core-sheath bicomponent fiber, (**c**,**d**) before and after melting of the PET/PTT side-by-side bicomponent fiber, (**e**,**f**) are images of PET/PA6 fibril matrix bicomponent fiber before and after melting, (**g**,**h**) were images of PET/PA66 lobe bicomponent fiber before and after melting.

**Figure 4 polymers-15-00842-f004:**
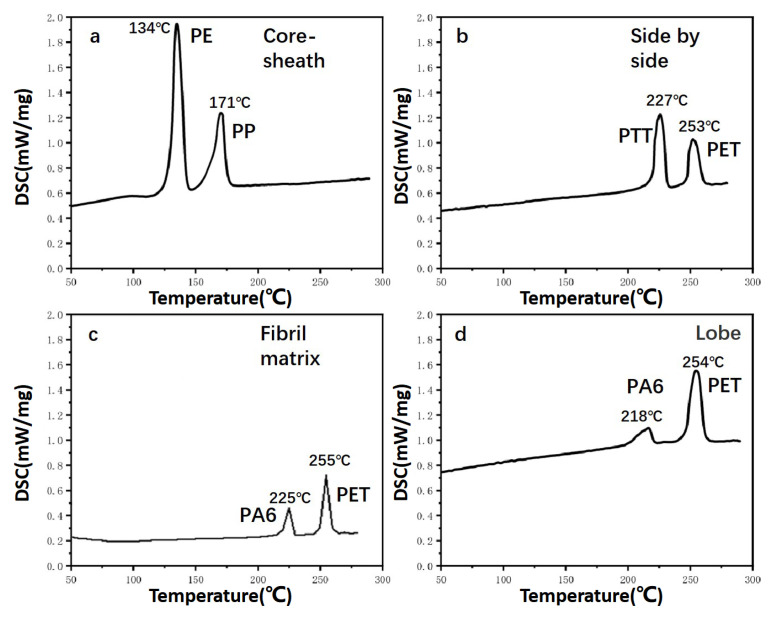
DSC diagram of four kinds of bicomponent fibers: (**a**) core-sheath bicomponent fiber, (**b**) side-by-side bicomponent fiber, (**c**) fibril matrix bicomponent fiber, and (**d**) lobe bicomponent fiber.

**Figure 5 polymers-15-00842-f005:**
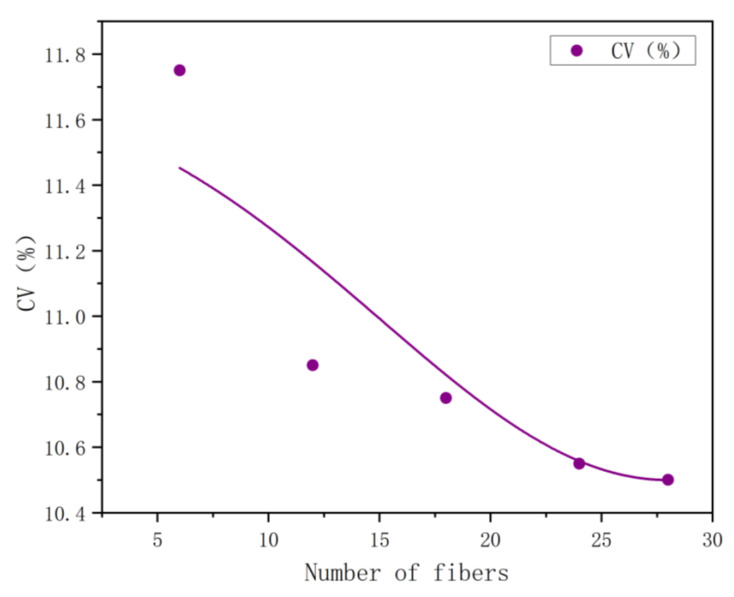
Trend graph of the *CV* value of side-by-side bicomponent fiber with the number of testing specimens.

**Figure 6 polymers-15-00842-f006:**
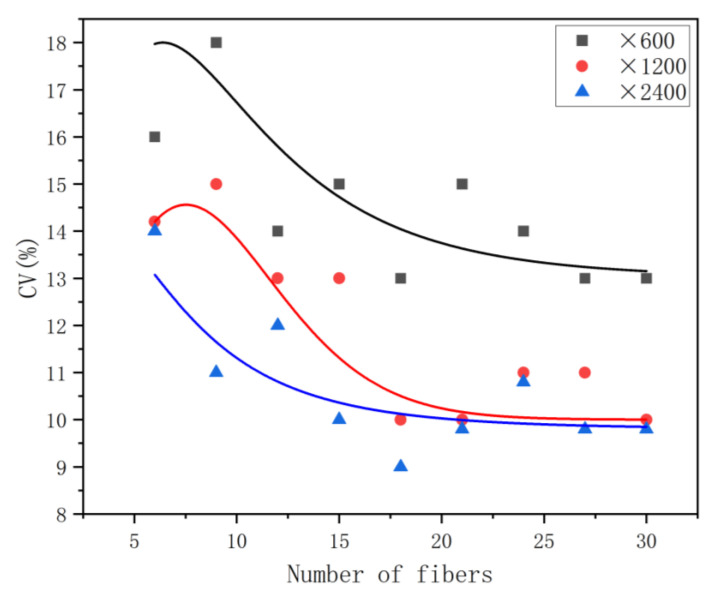
Influence of magnification on the *CV* Value of side-by-side bicomponent fibers.

**Figure 7 polymers-15-00842-f007:**
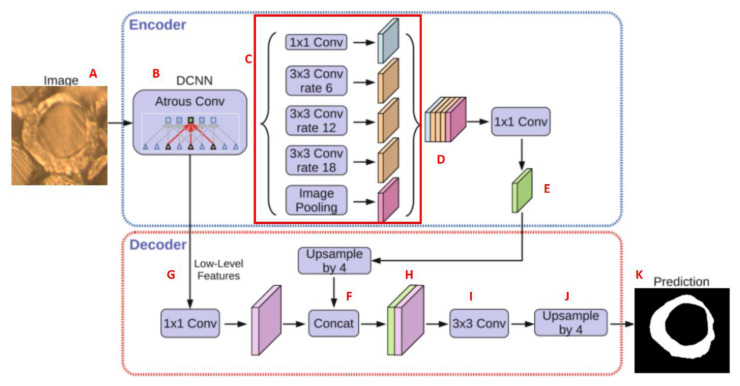
Establishment of DeepLabV3+ model.

**Figure 8 polymers-15-00842-f008:**
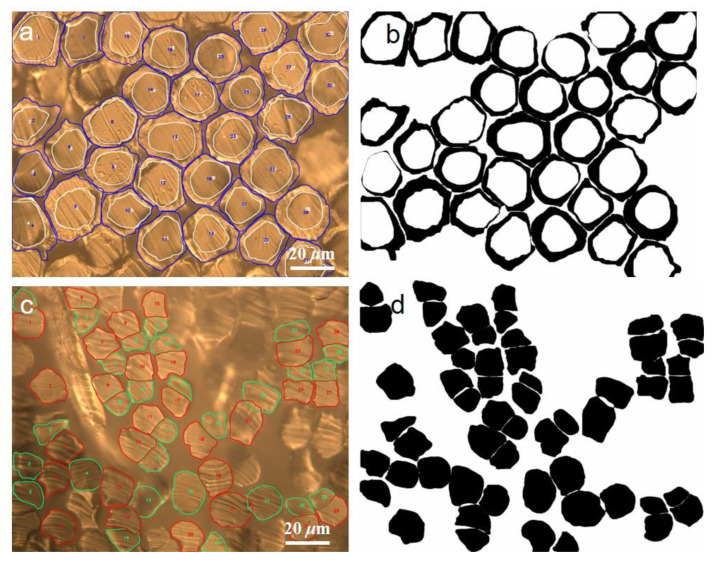
Recognition effect of the image recognition model on each component in bicomponent fiber. (**a**,**b**) images showed the effect of identification of the sheath and core layers by recognition models,(**c**,**d**) showed the effects of image recognition models on identifying each component of side-by-side bicomponent fibers.

**Table 1 polymers-15-00842-t001:** Allowable deviation rate of linear density of the chemical fiber in relevant standards (*E*, %).

Reference Standards	Premium Grade Products	First Grade Products	Qualified Products
FZ/T 52037-2014 Sea-island polyester and polyamide bicomponent staple fiber	±3	±4	±8
FZ/T 52034-2014 Polyethylene/polyethylene terephthalate (PE/PET) bicomponent staple fiber	±6	±8	±10
FZ/T 52024-2012 Polyethylene/polypropylene (PE/PP) bicomponent staple fiber	±6	±8	±10
ASTM D2497-2018 Standard Tolerances for Manufactured Organic-Base Filament Single Yarns	≥4.4 tex or 40 denier, ±4; <4.4 tex or 40 denier, ±6.0.		

**Table 2 polymers-15-00842-t002:** Functional relationship between *N* and *t* (95% confidence level).

Testing Specimen Number	30	30–40	41–60	61–120	121–230
confidence coefficient t	2.04	2.03	2.01	1.99	1.97

## Data Availability

The data presented in this study are available on request from the corresponding authors.

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
