# Peer review of "A Rapid Quantitative Analysis of Bicomponent Fibers Based on Cross-Sectional In-Situ Observation"

_polymers, 2023, doi:10.3390/polym15040842_

Round 1

Reviewer 1 Report

The present manuscript, “A rapid in-situ quantitative analysis of bicomponent fibers,” is too

lengthy, and the statements given in the abstract are not reflected in the result and discussion

part. Thus the manuscript needs proper modifications based on the comments given below:

Abstract: line 18-20

How the authors can justify that;

i). The time of the material analysis is 10 min.

ii). The total cost of the material analysis is only 10000 dollars.

iii). only half an hour of training is required for the new person who is not aware of the

instrument to perform the whole analysis

Line 12-14

The author’s have used the word in-situ (in line 12) while in situ (in line 14), similarly this has been repeated many times in whole manuscript. Kindly rectify this, and use similar way of writing the words.

Line 17

Write full form of “DSC”.

Introduction:

Line 46

Explain PET and PPT, similarly other abbreviations used in the manuscript

Line 50------- non-embedded fiber cross sections was developed in this work.

Was this facility not available in their lab before performing the present study?

Result and Discussion:

Line 101. What is the meaning of a non-embedded preparation method?

Line 108-9

How the transmittance light was measured, and how much is the difference in the transmittance?

value (write some number) to distinguish between the natural and synthetic fibers?

Line 128-129

Formic acid dissolved the package components, so it is a nondestructive technique because, after

analysis, the fiber cannot be used for further applications. Thus the testing is not practical.

Line 148-165

In Figure 3, according to the author’s reports it’s reflected that, when the average temperature increases then standard deviation increases also the density is increases except for PP core-sheath bicomponent fiber. How the authors have calculated these three parameters and what is the relation between them?

Line 151-152

------ the sheath layer was determined to be a PE layer with a density of 0.95 g/cm3.

How the authors obtained this (0.95 g/cm 3) number?

Line 156-166

All these methods are not fast as claimed in the abstract. Also, they are destructive because the

analyzed materials cannot be used further for practical applications after analysis. Kindly

explain

Line 184-186

In Figure 5, the DSC value is more at low melting point of core-sheath and side-by-side bicomponent fiber while the DSC value is less at low melting point of fibril matrix and lobe bicomponent fiber. Kindly explain it.

Line 241-243

The plot of Figure 6 needs to redraw just by plotting the smooth curve fitting of it at the place of joining of data point.

Line 254-256

In Figure 7, why the number of data points is less for magnification 1200 in comparison to 600 and 2400?

Line 264-265

Is there any limit for the higher diameter for proper distinction? What is the average diameter of

the synthetic and natural fiber so that by measuring the diameter of the unknown fiber, it can be

classified/categorized as synthetic or natural fiber?

Line 275-276

What is the basis of the selection of the “default value of the confidence coefficient t as 2.03”?

Line 310-311

------- and the value of this system was not more than $10000

How the authors can justify this as for the preparation of the specimen to be used in the

microscope, a lot of recurring costs of chemicals and laborers are required.

Reviewer 2 Report

The manuscript of Qin et al., developed a rapid in-situ quantitative analysis method for bicomponent fibers. The approach of the authors is well performed, the article is well written and the applied methodology merits publication in Polymers. After reading the manuscript I have minor/major comments which ought to be addressed before publication.

-        - Some of the mentioned references are extremely difficult to track. For instance reference 21 and 22 (but not limited to these examples). Even through the advanced search function of the journals website the reviewer was not able to access these publications. Can the authors provide a direct link to these papers? For this reason the self-citation criteria could not be checked.

-        - Suggestion; remove “them” in Line 133.

-        - Please add in Figure 1,2,3 and 4 which microscope that was used for the different microscopic images.

-       -  Please place “3” in Lines 152, 155, 160, 163, 174, 177, 180 and 183 in superscript.

-     -    Missing space in Lines 145 and 168.

-        - Please increase the quality of Figure 5, increase font size of axes titles and numbers to facilitate reading.

-       -  What are the lines in Figure 7? Is this some sort of regression ?

-      -   Please introduce a reference for following sentence “The diameter uniformity of the synthetic fibers was higher than that of natural short fibers, such as cotton, hemp, and wool.” Lines 264-266.

-       -  The section “Determination of Allowable Error Rate E” is somewhat unclear. 1/ What do the authors mean with that the diameter uniformity was determined by the management level ? 2/ are the values for E in Table 1 experimentally determined or are the derived from the literature ? If the latter, please add reference.

-      -  The authors mention several times that the cost of this analysis is approximately $10 000. In Lines 308-311 it is mentioned that the value of the system (microscope?) is $10 000. Do the authors mean that the cost of the microscope is $10 000 or the cost of 1 complete analysis is $10 000 ?  As the authors mention several times the economic benefit of this method, this should be elaborated.

-       -  In the title in-situ is written as “in-situ” throughout the text “in situ” is used. Please be consistent throughout the text.

Round 2

Reviewer 1 Report

None

Reviewer 2 Report

The reviewers have revised the manuscript accordingly, and therefore the reviewer accepts this manuscript for publication in Polymers.